# OpenReview forum: "Quantifying Social Biases Using Templates is Unreliable"
_NeurIPS.cc/2022/Workshop/TSRML — TSRML2022_

### Official Review · Reviewer_g9nX · 2022-10-10
**Addresses an important problem, but lacks clarity and sufficient justification of novelty**

**Overall Rating:** 6

**Summary:**

This paper examines the brittleness of LLM template-based bias measurements to the choice of template. The authors find that bias measurements vary considerably across semantically-equivalent templates, calling for more caution when evaluating model bias.

**Strengths:**

Quality:
- excellent exposition of bias evaluation, with lots of relevant literature
- authors observe significant fluctuations in bias measurements after applying semantics-preserving transformations to templates
- well-explained and sensible experiments with bias evaluation benchmarks across four different tasks
- good calls to action (e.g., more human-crafted bias evaluation datasets, diverse templates)

Significance:
- addresses limitations of bias evaluation benchmarks, which is critical to robustly and validly measuring model biases

**Weaknesses:**

Quality:
- authors should further examine template fillers (e.g., [GENDERED WORD], [TARGET])
  - [1] found that sentences (after filling templates) are often grammatically incorrect
  - how grammatically correct are transformed sentences after filling templates? how much could this contribute to the observed changes in bias measurements?

Originality:
- novelty of template transformations (especially compared to [2]) needs to be further justified

Clarity:
- please include metric formulas for all the tasks; written descriptions of metrics are unclear at times

[1] https://aclanthology.org/2021.acl-long.81/
[2] https://arxiv.org/pdf/2112.07447.pdf

**Overall Recommendation:**

Work addressing an important problem, but improved clarity and further justification of novelty are required.

**Review Confidence:**

4: The reviewer is confident but not absolutely certain that the evaluation is correct

---

### Official Review · Reviewer_reyw · 2022-10-19

**Overall Rating:** 7

**Summary:**

The paper considers testing and quantifying large language model biases using templates. The authors investigate 4 NLP tasks: sentiment analysis, toxicity detection, natural language inference (NLI) and masked language modeling (MLM). They manually modify (in a semantically preserving manner) templates that have been proposed in previous works and were used to quantify bias for these tasks. They find that bias mesurements vary broadly across the template modifications (ranging from 33% difference for sentiment analysis to 162% difference for MLM).

**Strengths:**

* The paper is extremely clear and well-written.
* The considered problem is of high importance.
* The experimental evaluation is extensive, covering 4 NLP tasks.
* The experimental results support the main claim made by the paper.
* The authors propose a way going forward of how to deal with the variations of the results obtained from different/modified templates:
	* Considering more test cases (as in HateCheck [Rottger et al., 2021])
	* Automatic generation of more diverse and natural sets of instances (as in AdaTest [Ribeiro and Lundberg, 2022])

**Weaknesses:**

* I believe the numbers in the first two rows do not match those in Table 8 (3 + 2 = 5 in row1, Table 2 vs 6 modified templates in Table 8, and 7 vs 6 respectively for row 2)
* I have a question/slight concern about the NLP experts and the filtering of the templates. How many experts were employed and how were they selected? Is there any danger of implicit bias when filtering the templates according to their decisions?

Questions/suggestions for future work:
* Is there any way to reliably scale (and even automate, partially at least) the process of template generation? I believe AdaTest is in that direction.
* How can we be confident (if ever at all) in the final selection of tests and templates, or the authors envision continuous generation and search for model defects.

**Overall Recommendation:**

The paper is well-written and presents extensive evaluation supporting the claim that quantifying bias with (single, individual) templates is noisy and can produce very different results (when the templates are modified). Therefore, I recommend acceptance.

**Review Confidence:**

4: The reviewer is confident but not absolutely certain that the evaluation is correct

---

### Decision · Program_Chairs · 2022-10-23

**Decision:**

Accept

**Comment:**

Good work on identifying problems in existing bias evaluation methods with templates.